# Multiple Chemical Sensitivity

**DOI:** 10.3390/brainsci12010046

**Published:** 2021-12-29

**Authors:** Gesualdo M. Zucco, Richard L. Doty

**Affiliations:** 1Department of Philosophy, Sociology, Education and Applied Psychology, University of Padova, 35100 Padova, Italy; 2Smell and Taste Center, Department of Otorhinolaryngology, Head and Neck Surgery, Perelman School of Medicine, University of Pennsylvania, Philadelphia, PA 19104, USA; richard.doty@uphs.upenn.edu

**Keywords:** multiple chemical sensitivity, chemical intolerance, toxicant induced loss of tolerance, prevalence, etiologies, assessment, olfaction, psychopathology

## Abstract

Multiple Chemical Sensitivity (MCS), a condition also known as Chemical Sensitivity (CS), Chemical Intolerance (CI), Idiopathic Environmental Illness (IEI) and Toxicant Induced Loss of Tolerance (TILT), is an acquired multifactorial syndrome characterized by a recurrent set of debilitating symptoms. The symptoms of this controversial disorder are reported to be induced by environmental chemicals at doses far below those usually harmful to most persons. They involve a large spectrum of organ systems and typically disappear when the environmental chemicals are removed. However, no clear link has emerged among self-reported MCS symptoms and widely accepted objective measures of physiological dysfunction, and no clear dose-response relationship between exposure and symptom reactions has been observed. In addition, the underlying etiology and pathogenic processes of the disorder remain unknown and disputed, although biologic and psychologic hypotheses abound. It is currently debated whether MCS should be considered a clinical entity at all. Nevertheless, in the last few decades MCS has received considerable scientific and governmental attention in light of the many persons reporting this illness. In this review, we provide a general overview of the history, definition, demographics, prevalence, and etiologic challenges in defining and understanding MCS.

## 1. Introduction

Multiple Chemical Sensitivity (MCS) is a controversial disorder in which patients claim to become ill from environmental exposure to low levels of largely petroleum-based and often unrelated chemicals, including cleaning products, detergents, diesel exhaust, formaldehyde, plastics, carpets, epoxy, pesticides, and some synthetic and natural fragrances [1,2,3,4,5,6,7,8,9,10,11]. The symptoms can arise gradually over time from repeated low-level chemical exposures or, in some instances, immediately following a single unexpectedly high-level exposure. Often MCS is assumed to follow two distinct phases, ‘sensitization’ and ‘triggering’ [12]. In the first phase, symptoms appear in response to a chemical exposure episode or episodes which commonly cause aversion to the stimuli in symptomatic persons. In the second phase, symptoms are generalized (a spreading effect) to a larger set of odoriferous substances or chemicals. In this phase when a body organ becomes involved, other organs subsequently follow [13]. Among its symptoms, which can occur in combination and can encompass multiple organ systems, are perceived hypersensitivity to the smell of chemicals, nausea, dizziness, headache, upper respiratory discomfort, runny eyes, chest and throat pain, arthralgia, dyspepsia, fatigue, lack of concentration, memory difficulties, depression, anxiety, mood disruption, and a range of other cognitive and psychological disturbances [2,5]. Removal from the environment of the offending agents frequently mitigates the experienced symptoms.

At present, no standard medical tests have been identified that correlate with the numerous multisystem complaints of MCS patients. The underlying causes of the syndrome, whether biologic or psychologic, are not fully understood [6,7,12,14,15,16,17,18,19,20,21,22]. Indeed, the elusiveness of the syndrome and the heterogeneity of the symptoms cast doubt as to whether MCS is an actual unitary clinical entity. Not surprising, MCS is often juxtaposed to, and sometimes combined with, other diseases for which definitive diagnoses and explanations are also found wanting, including fibromyalgia (FM), Gulf War syndrome (GWS), chronic fatigue syndrome (CFS), sick building syndrome (SBS), and electromagnetic radiation exposure (ERE) [18,23,24,25,26,27,28,29,30,31,32].

In this review we provide a general overview of the history, definition, demographics, prevalence, and etiologic challenges, both biologic and psychologic, in understanding MCS. While every effort has been made to address, at least on a summary basis, all major aspects of MCS, a complete rendition of this large literature is not possible in one article. The reader is referred elsewhere for other contemporary reviews that encompass aspects of this intriguing but largely enigmatic disorder [6,7,9,16,17,18,19,33,34,35,36,37].

## 2. History of the MCS Concept

Interest in chemical sensitivity and environmental illness–interest that forms the beginnings of the MCS movement–became popular in the 1950s, when the allergist Theron Randolph (1906–1995) argued that allergic reactions from exposures to environmental chemicals, pollutants, and several kinds of foods were responsible for a large range physical and psychological diseases. Randolph [38,39,40] coined the phrases *food addiction* and *ecological mental illness* to stress the toxic role of allergens in provoking hypersensitivities and abnormal immune responses. He is also recognized as the founder of the controversial medical field he termed *Clinical Ecology* [40,41], whose tenets, diagnostic methods, and treatments generally fall outside standard scientific methodology. The ideas related to this pathogenic model revolve around concepts that have not been tested in the peer-reviewed medical or scientific literature. These include food addiction and allergy, specific adaptation, total body load, biochemical overloading, spreading phenomena, susceptibility, and sensitivity [42].

The approach taken by Randolph and his followers to the etiology and treatment of the wide variety of purportedly environmentally caused diseases and disorders has been considered, in light of lack of scientific evidence, to be pseudoscience by many within the medical community. Thus, MCS has yet to be accepted as a disease entity by such authoritative medical organizations as: the American Academy of Allergy [43,44,45], the American College of Physicians [46], the American College of Occupational and Environmental Medicine [47,48], the American Council on Science and Health [49], the American Medical Association [50,51], the Royal College of Physicians and Royal College of Pathologists [12,42,52,53,54,55]. Despite such repudiation, the term Clinical Ecology has received new impetus in medical movements that stress inter-relationships between the individual, his or her microbiome, and the wide range of social, political, and economic ecosystems that determine an individual’s health [56]. Moreover, as noted later in this review, a number of governmental agencies have accepted MCS, as defined by other classification systems and empirical studies, implying the nominal existence of such a disorder.

### Operationally Defining MCS 

More orthodox support for MCS as a specific diagnostic entity was provided in the 1980s by Cullen [57] who established the first clear-cut operational criteria for case definition (see also [58,59]). Since that time, other criteria have been proposed and numerous terms have been introduced to operationally redefine MCS. Among such terms are Chemical Intolerance (CI), Chemical Sensitivity (CS), Idiopathic Environmental Intolerance (IEI), Toxicant Induced Loss of Tolerance (TILT), Total Allergy Syndrome (also termed Twenty Century Disease), Chemical Injury, Chemophobia, Toxic Injury, Environmental Hypersensitivity Syndrome (EHS), and Environmental Illness. This plethora of terminology reflects, in large part, the difficulty of finding consensual operational definitions and acknowledged causes of the symptoms ([15,16,18,19,20,55,57,60,61,62,63,64,65].

Cullen’s [57] operational and definitional criteria for MCS, were as follows: (a) the presence of an acquired disorder following documentable environmental exposure to chemicals or toxins; (b) symptoms involving more than one organ; (c) the occurrence of symptoms as a response to predictable different classes of chemicals or odors; (d) doses not harmful to most persons; and (e) the absence of a correlation between the presence of the symptoms and objective routine medical tests. According to Cullen, symptoms may arise following a single high-level and often accidental exposure to a harmful environmental substance or substances (e.g., pesticides, gasoline, organic solvents, organophosphates, pyrethrums) or from repeated and continued low level exposures to such substances. Intoxication, allergies, and other pathologies with acknowledged causes were considered exclusion conditions for the MCS diagnosis.

Cullen’s defining criteria were well received by many in the scientific community and were incorporated within the United States MCS consensus criteria [64]. It was concluded that MCS is a chronic condition, with symptoms reproducible after repeated exposures and that it improves or even resolves when the triggering incitants are removed (see, also [61]). In addition to establishing a differential diagnosis between MCS and other diseases (e.g., cardiovascular, gastroenterologic, psychiatric, and neurotoxicological), researchers and clinicians who reached the aforementioned diagnostic criteria suggested, as a first step, the use of a questionnaire devised by [66,67]. This questionnaire, termed the *Environmental Exposure and Sensitivity Inventory* (EESI) is designed to detect people sensitive to common chemical triggers.

More recently, Lacour et al. [5] validated, revised, and extended Cullen’s case diagnostic criteria, providing further guidelines, namely: (a) that self-reported odor sensitivity symptoms should primarily involve the central nervous system (the so-called non-specific complaints, like headache); (b) such symptoms can secondarily adversely affect other organs (the so-called optional complaints, like gastrointestinal, musculoskeletal, dermatological, and related complaints); (c) the symptoms should persist at least for 6 months (to rule out persons with self-limited acute or subacute toxic reactions); and (d) the symptoms should adversely impact social and occupational lifestyles. Moreover, medically unexplained syndromes which overlap MCS, such as CFS and primary FM, should be considered exclusion criterion regardless of whether they appear to be a direct consequence of MCS.

Presently the diagnostic criteria of Cullen and Lacour et al. are the ones that are most widely employed, although alternatives have been suggested ([18]; see also [67,68,69,70,71]). For example, Ishibashi et al. [72] proposed a classification system of MCS and SBS that focuses on actual symptoms and disease pathogenesis, rather than on general criteria. They divide MCS into four general classification categories based upon symptoms that arise from (a) chemical intoxication, (b) specific chemical exposures (e.g., to a new house environment), (c) psychological factors, and (d) allergies.

## 3. Olfactory System Involvement in MCS

People afflicted with MCS very often report a heightened sensitivity to odorants. Some report being able to smell odors that others cannot. However, studies employing psychophysical methods have been unable to document enhanced olfactory sensory sensitivity, as measured by odorant thresholds in such persons. For example, Doty et al. ([73], see also [74]) found no evidence that the detection thresholds for two target stimuli, phenyl ethyl alcohol and ethyl methyl ketone, were any different in 18 MCS subjects than 18 matched controls, despite the self-reported hypersensitivity by the MCS subjects. An association was documented, however, between MCS and increased nasal airflow resistance, respiration rate, and heart rate. Similar failures to see differences in olfactory thresholds or other olfactory measures (e.g., ratings of odor intensity, odor identification) between MCS patients and controls have been reported by others [8,75,76,77,78,79]. Hyper-reactivity, rather than hypersensitivity, may explain some reports of heightened odor sensitivity by MCS patients. Moreover, a broader range of olfactory stimuli may be needed to detect threshold differences between MCS patients and controls.

That being said, odorants are known to provoke lower and upper airway irritation and inflammation in some persons. This may be interpreted by them as olfactory hypersensitivity, even though the olfactory system, per se, is not directly involved in the pathogenesis or mediation of the sensations. The coincidence of the perception of an odor with such sensations would facilitate this attribution. Indeed, depending upon the concentration and duration of exposure, nearly all odorant chemicals can produce some intranasal sensations via trigeminal nerve nociceptors [73]. Moreover, vagal nerve nociceptors detect some volatiles that reach the throat and upper airways [80]. Airborne chemicals have rather direct access to such nociceptors, unlike those in the skin, since the free nerve endings are apically located in the mucosa and lack the cover of squamous epithelium [81]. Trigeminal reactivity would explain the Doty et al. [73] findings of increased nasal airflow resistance, respiration rate, and heart rate in MCS patients, since these types of reflexes are dependent upon the trigeminal system. Volatile chemicals can not only affect respiration, but can cause laryngeal symptoms which, in the extreme, can induce vocal cord dysfunction [82]. In accord with this concept is evidence that a number of patients who complain of chemically induced upper and lower airway symptoms have lower thresholds for coughing and other symptoms—symptoms that can be induced by chemical activation of the vagus nerve. It is of interest that MCS patients, relative to controls, have a hyper-reactive cough response to capsaicin inhalation challenge which is associated with greater increases in nasal lavage concentrations of nerve growth factor (NGF). NGF is a key neurotrophin found in the serum, bronchial tissue, and broncho-alveolar fluid of patients with allergy, asthma, and allergic rhinitis [83]. It should be noted that number of the MCS symptoms overlap with those of other hypersensitivity syndromes such as asthma, migraine and urticaria [84].

## 4. Demographics of MCS Patients 

The emerging profile of people afflicted by MCS finds them, on average, to be well-educated, of higher socioeconomic status, and middle aged [16,18,85,86,87]. Compared to younger persons, those 65 years of age and older are less likely to identify themselves as chemically sensitive [88]. Like the sick building syndrome [89], there is a remarkably higher prevalence in women than in men [16,18,21,90,91], with the percentage of women ranging from 60% to 88% [18,57,92,93,94]. However, exceptions do occur (e.g., [95]).

The reasons for demographic differences between MCS and non-MCS subjects is unclear. In some cases, sampling issues may be involved. For example, it has been found that volunteers typically have more education, higher occupational status, earlier birth position, lower chronological age, higher need for approval, and lower authoritarianism than non-volunteers [96]. Women are more likely to volunteer for experiments, a phenomenon that is also influenced by their endocrine state [97]. Other potential non-mutually exclusive explanations for the higher number of women in MCS cohorts are that women pay more attention than men to external cues to define their symptoms, may spend more time in poorly ventilated homes, and are more prone to the psychological and somatic disorders associated with MCS [16,91,98,99,100]. Experience with odorants also could be involved since repeated exposure influences sensitivity to some odorants [101,102,103], a phenomenon that is stronger in women than in men [104].

## 5. Prevalence of MCS

Accurate assessments of MCS prevalence are difficult to establish and substantial differences exist among studies. This may be due to the elusiveness of the syndrome (i.e., the difficulty of establishing consensus on etiology, diagnosis, and pathogenesis), its apparent similarity to other diseases that hinder a distinctive differentiation (e.g., FM, CFS, SBS, Idiopathic Intracranial Hypertension, IIH), and the context in which data are collected. Prevalence rates differ between (a) clinical settings where patients are diagnosed by a medical professional, queried about symptoms, and alternative diagnoses provided and (b surveys in which self-reported symptoms are not confirmed professionally. Additionally, the specifics of the questions that are asked can also impact estimates of prevalence (e.g., by asking only about physician-diagnosed MCS). As can be seen from Table 1, prevalence rates are typically lower when patients are evaluated medically and can vary dramatically among studies, ranging from <1% to 33%. As noted by Herr et al. [105], the majority of patients who attribute their illness to environmental causes can actually be assigned to other clinical diagnoses, including respiratory diseases, skin diseases, and gastrointestinal diseases, resulting in the identification of no more than 15% of persons with symptoms suspected to reflect environmental causes. Remarkably, somatization disorders can be diagnosed in 40% to 75% of patients with environmental complaints.

## 6. Acceptance of MCS as a Disorder by Governmental Agencies

A number of countries and health care communities have recognized MCS or similar entities as a debilitating illness. Germany first recognized MCS in 1998, followed by Denmark, Austria, Luxemburg, Spain, and Finland. MCS is classified in Germany and Austria under the ICD-10 code T78.4 (unspecified allergies, Nitrous Oxide System-hypersensitivity, NOS-idiosyncrasy). In Sweden, Electro-Magnetic Hypersensitivity (EMH) is recognized. Each of these countries have assigned appropriate billing codes to include the disorder within their health care systems. However, the World Health Organization (WHO) has not assigned any clear separate code for MCS or related diseases within the International Classification of Diseases (ICD-10). Two existing generic codes, J68.A and T78.4 (which respectively refer to “unspecified respiratory condition due to inhalation of chemicals” and to “unspecified allergies”) might be applied to some MCS cases, although they do not specifically include MCS symptoms. In the United States, MCS has been partially recognized by several medical authorities. For example, the Centers for Disease Control and Prevention (CDC) established the presence of chemical intolerances in patients affected by CFS and Myalgic Encephalomyelitis (ME). Federal agencies such as the Environmental Protection Agency (EPA), the Department of Housing and Urban Development (HUD), and the Social Security Administration (SAA), as well as legislation such the American with Disabilities Act (ADA), have paid at least some attention to MCS [34,120]. Japan, which has acknowledged SBS for a number of years [72], also officially acknowledged MCS in its health system ICD-10, code T65.9 (unspecified respiratory conditions due to inhalation of fumes, gas, and chemical vapors [72]). The Canadian Federal Centre for Occupational Health and Safety, as well as some local Canadian agencies, has recognized MCS and other related syndromes like SBS as pivotal health and safety issues [18]. In Italy, MCS is not yet nationally recognized by the Ministry of Health, but it is recognized locally by some regional authorities as a rare disease. Moreover, a bill was introduced a few years ago to the Italian Parliament entitled “Provisions in favor of individuals with multiple chemical sensitivities,” although its passage has not occurred [121].

## 7. Etiology of MCS 

Although the etiology of MCS remains disputed and its mechanisms generally undefined, the main explanatory categories fall into two broad categories: psychological and biological. A psychological basis for many if not most of the symptoms of MCS patients arises from a large literature, described in detail in the next section. Many such symptoms are the same symptoms seen in a wide range of psychological disorders. Often MCS patients, when medically evaluated, end up with non-MCS diagnoses. Most salient is the fact that such symptoms can be conditioned to environmental agents, such as odorants, in accord with traditional theories of learning [122].

### 7.1. Psychological Theories of MCS

Support for the psychogenic hypothesis of MCS comes from multiple sources. High levels of depression, anxiety, and mental distress have been noted in MCS samples, including alterations in cognitive and emotional processing [123,124,125,126,127,128,129,130,131,132,133]. In a Canadian survey involving 21,997 citizens ranging in age from 15 to over 80 years of age, higher scores for major depressive disorders, mental distress, and generalized anxiety disorder were found in the MCS patients than in controls [134]. The authors hypothesized that MCS patients have a greater sensitization towards environmental stimuli on which they focus their attention to explain their psychological symptoms. Similar results were found in a previous large Canadian survey by Park and Knudson [114].

Such findings, and the lack of uniform biologic explanations for the disorder, have led some to conclude that all MCS cases are likely psychogenic. Regardless, it remains a matter of debate as to whether the psychological symptoms are a causative factor or a consequence of the disorder. Labarge and McCaffrey [16] have suggested that, in most cases, a previous history of mental disease (e.g., anxiety, depression, mood disorders) exists prior the diagnosis of MCS (see also [124,135,136,137]). In contrast, others have suggested that the environmental sensitivity pathology is more likely the cause of the psychological disturbances, predating the psychological symptoms [2,18,138,139]. Such symptoms strongly impact the whole life of MCS patients, including their social relationships, personal concerns, jobs, and quality of life [140,141].

Described below are examples of psychiatric disorders which share many of the same symptoms or features of MCS. These examples are followed by information on how cognitive factors, as well as classical conditioning, can result in both perceived and actual MCS-related symptomatology. In classical conditioning (CC), for example, external stimuli, such as odorants or airborne irritants, can become associated with emotional events which, in turn, produce MCS symptoms.

### 7.2. Panic and Post-Traumatic Stress Disorder Symptoms 

Many MCS-related symptoms are similar to those of classic panic disorders, including hyperventilation, dizziness, palpations, chest pain, nausea, sweating, and other autonomic nervous system responses. Such symptoms, when induced experimentally in provocative challenges by intravenous sodium lactate [142,143], prove to be greater in MCS patients than in controls [144,145]. Similar reactions to inhalation challenges from organic solvents were described by Leznoff [146] and by Dager et al. [147]. According to Tarlo et al [138], panic responses in MCS patients may reasonably be the expression of somatic symptoms often attributed to toxic mechanisms.

Stenn and Binkley [148] have argued that besides being conceptualized as a panic disorder, MCS shares characteristics of a post-traumatic stress disorder in which the past traumatic experience is triggered by an exposure to an odorant. Thus, a re-experience of the odorant provokes the onset of symptoms. Other authors also suggest, in light of their clinical similarities, that a similar underlying mechanism can be the unifying factor of MCS, CFS, SBS, Burnout syndrome (BS) and Candida Syndrome (CS) within the field of somatoform disorders [25,87]; see also [149] for findings supporting a functional somatic syndrome in patients with indoor air related intolerance). Park and Knudson [115] provide an interesting report on the similarities of the medically unexplained physical symptoms MCS, CFS and FM.

### 7.3. Somatization Disorder Symptoms

Patients with somatization disorders are characterized by having excessive thoughts, feelings and behaviors related to physical symptoms such as weakness, pain, or shortness of breath. According to the American Psychiatric Association (APA), the physical symptoms may or may not be associated with a diagnosed medical condition, but the patients believe they are sick and are not malingering. According to the APA, the diagnostic criteria are “excessive thoughts, feelings or behaviors related to the physical symptoms or health concerns with at least one of the following: (1) ongoing thoughts that are out of proportion with the seriousness of symptoms; (2) ongoing high level of anxiety about health or symptoms; (3) excessive time and energy spent on the symptoms or health concerns; (4) at least one symptom is constantly present, although there may be different symptoms and symptoms may come and go” (www.psychiatry.org, accessed on 13 October 2021).

As noted earlier in this review, somatic symptoms such as pain, shortness of breath, excessive sweating, confusion, and tachypnea are common in many MCS patients [108]. Such sensations frequently are accompanied by such psychological disturbances as depression, obsessive behavior, and anxiety [87,92,114,125,131,137,141,142,150,151,152]. However, a minority of studies have not found evidence for such somatization (e.g., [153]).

### 7.4. Psychological Beliefs and Expectancies

Cognitive factors which give rise to misinterpretation of the exhibited symptoms and therefore to perceptual biases can also drive MCS symptomatology. The influences of cognitive elaborative processes and attentional variables regarding how odorants are perceived psychologically is well established in healthy subjects and has been demonstrated in MCS patients. Zucco, Militello and Doty [8], for instance, described a procedure that proved useful in differentiating organic from psychological elements in a woman who met a strict set of criteria for MCS [57]. She was tested on two test occasions. On the first, she was found to have no olfactory dysfunction, as determined from standardized olfactory tests. On the second, a series of odorants, as well as a blank stimulus, was presented to her. On some trials she was informed the odorant she was receiving was harmless and on other trials it was harmful. Blood pressure and heart rate was measured before and after the presentation of each stimulus. The results clearly showed that potentially harmful odorants presented as harmless were judged significantly less intense, and triggered fewer symptoms, than harmless odorants presented as harmful. When an odorless stimulus was presented as harmful, the patient provided higher intensity evaluations and exhibited more symptoms than when it was presented as harmless. These phenomena were not present in three non-MCS controls. Clearly, the patient’s the patient’s symptoms were not due to the odorants, per se, but reflected cognitive, attentional, and emotional responses to what the odorants were believed to represent (see, also [154] on the related iatrogenic power of suggestion).

The procedure for differentiating organic from psychologic symptoms of MCS used in this study may have considerable value to other situations where MCS patients are being assessed. Its findings are in general agreement with those from similar studies of normal subjects [6,7,155,156,157,158,159,160]. For example, Dalton et al. [157] exposed 90 healthy subjects to 200 ppm phenyl ethyl alcohol (PEA, a relatively nonirritating rose-like smelling odorant) and to 800 ppm acetone. Subjects were assigned to three groups: the positive bias group was informed that they would be exposed to natural extracts commonly used in aroma therapy that may have beneficial effects on mood and health; the negative bias group was told they would be exposed to industrial solvents that purportedly caused health effects and cognitive problems; and the neutral bias group was told they would be exposed to standard odorants commonly used in olfactory research. Acetone and PEA detection thresholds were equivalent before and after the test exposures. However, for both acetone and PEA, the negative bias group rated the stimuli as more intense and, on average, producing the most irritation, as well as the most health-related symptoms after chemical exposure compared to the positive bias group. According to the authors, self-generated symptoms may be activated by personal mental schemas, as it may happen for people living close to hazardous places, which are considered toxic, even in absence of objective measures (see also [78]).

### 7.5. Classical Conditioning

Classical conditioning (CC) is one of the most investigated underlying putative psychological mechanisms of MCS. In such conditioning, a neutral stimulus takes on the traits of another stimulus after it is repeatedly paired with that stimulus [161,162,163,164,165,166] The pioneering description of CC comes from the Russian physiologist, Ivan P. Pavlov, who repeatedly paired a sound, the neutral stimulus (NS), with food that induced salivation (the unconditioned response or UR). Later presentations of the sound alone (now termed the conditioned stimulus, or CS) induced salivation in the absence of the food. The salivation induced by the sound alone is termed the conditioned response (CR).

There are numerous examples of where CC can play a role in the emergence and maintenance of different kinds of somatic diseases, including symptoms associated with MCS [8,119,167,168]. When a stressful task is performed with a low-level odorant in the background, the odorant, which was not consciously recognized by the subjects in the initial pairing, appears to be able to influence moods and attitudes when encountered at a later time [165,169]. Pairing an odorant with motion sickness, e.g., nausea and vomiting induced in a rotation chair, can result in the odor alone later provoking the sickness. Klosterhalfen et al. and Bolla-Wilson et al. [170,171] describe cases in which headaches, nausea, and pain in the extremities followed accidental exposures to such odorous toxic agents as insecticides or solvents. Subsequently, these same symptoms could be elicited by odorants unrelated to the original conditioning such as cigarette smoke or car fumes (see also [15,172]).

It is important to note that a common treatment for MCS sufferers is to avoid xenobiotic triggers in order to regain health. Unfortunately, in doing so they still continue to experience their symptoms since deconditioning has not occurred. It is well known that a conditioned response can be maintained by operant conditioning via avoidance and escape behaviors (see, e.g. [173]). Of course, according to the CC model, anticipatory anxiety responses can be present in the patients (through self-generated mental cues or imaging) prior to the avoidance [174,175].

## 8. Biological Theories of MCS

A number of biological theories of MCS have been proposed, as described below. These theories overlap in many cases and therefore cannot be considered mutually exclusive. This is particularly true with genetic hypotheses that can involve numerous mechanisms.

### 8.1. Neurogenic Inflammation 

The neurogenic inflammation hypothesis suggests that odorous chemicals trigger the responses of unmyelinated c-fiber neurons which are widely distributed in the respiratory mucosa ([176,177,178,179]; see also [180]). This, in turn, results in the release of substance P, a key element of a number of inflammatory processes. The consequent immune response is then suggested to provoke central nervous system (CNS)-mediated symptoms such as emesis, nausea, mood disorders, and stress. Meggs [178], the originator of this hypothesis, introduced the term neurogenic switching which suggests that chemical stimulation at one body site (e.g., the mucous membrane of the nose) can lead, via the CNS, to inflammation in other distant sites, thereby provoking symptoms like headache or tachycardia. Related hypotheses can be found in Bascom [1] (see also [181]), who postulated that MCS is due to the expression of an amplified immune responses triggered by c-fibers and/or an altered function in the respiratory epithelium (see section on Immune System Dysregulation below).

### 8.2. Limbic System Dysfunction

The Limbic System Dysfunction Hypothesis (LSDH) relies to a large degree on studies of kindling [182,183]. Kindling, which was first demonstrated in rats, refers to increased brain electrical response following repeated low-level electrical stimulations of limbic structures. This can lead to a permanent lowering of the seizure threshold. According to the LSDH, recurrent low-level intermittent exposure to chemicals could produce a similar phenomenon, thereby establishing a permanent pathological response. This process is embodied within Miller’s model of Toxicant Induced Loss of Tolerance ([4,20,63,184,185]. According to Miller, toxicants such as pesticides, solvents, and various hydrocarbons can reach the brain through peripheral receptors of the olfactory system, bypassing the blood-brain barrier and directly accessing the limbic system (see [186] for a review of such transit). This is hypothesized as occurring in the absence of awareness and following brief exposures. Continued exposures to toxicants is suggested to make elements of the limbic system more sensitive to stressors. TILT, according to Miller [184], follows two steps. The first step is initiation, whereby people exhibit loss of prior tolerance after an acute or chronic exposure to chemicals. The second step is the triggering of symptoms, which occurs also in the presence of low quantities of the previous innocuous substances. Therefore, once a person develops chemical sensitivities, responses may occur to a broader range of xenobiotics at levels of exposure that were previously tolerated. TILT can explain the kind of symptoms characterizing some syndromes often juxtaposed to MCS, like SBS and GWS, the first being triggered by indoor volatile compounds (mold, formaldehyde, plasticizers, carpets, painting) and the second following a continue exposure to irritants like smoke from oil fires, diesel fuel, and pesticides. In accord with the limbic hypothesis, Heuser and Wu [187], using positron emission tomography (PET), found deep subcortical and limbic hypermetabolism in MCS patients, while Orriols et al. [181], using single photon emission computerized tomography (SPECT), found reduced inhibitory signaling from several olfactory areas (e.g., amygdala, hippocampus, right temporal cortex), suggesting heightened sensitivity in MCS patients. Similarly, Andersson et al., [188] observed limbic hyperactivity in an fMRI study of IEI patients exposed to low levels of olfactory and trigeminal stimuli. However, in contrast to these findings, two PET studies of 12 MCS patients found no meaningful resting functional imaging brain patterns specific to MCS [189,190,191].

### 8.3. Neural Sensitization and Hyperresponsivity 

This is a parallel hypothesis to LSDH proposed by Bell et al., [192]. Cases of neural sensitization (revealed by an increased EEG alpha frequency amplitudes) were observed after repeated intermittent exposures to chemicals in chemically sensitive women compared to normal controls [76]. Changes in skin conductance were also seen in MCS patients but not in controls [193]. Bell et al. [194] found that older persons who were intolerant to odors had higher heart rates and diastolic blood pressure across repeated tests than older persons who were more tolerant to odors, which is also consistent with predictions from the LSDH model for MCS.

### 8.4. Immune System Dysregulation 

Allergic responses and alterations in the immune system have been proposed as a possible etiological mechanism of MCS. However, sensitivity to chemicals cannot be considered an allergic response to pollutants and other environmental agents in the classical manner, since altered levels of circulating immunoglobulins and lymphocytes are not routinely observed [16]. Nevertheless, patients with documented measured exposure to molds have been shown to have abnormally elevated titers of antibodies to neural-specific antigens (e.g., immunoglobuline A, M and G) and to evidence various degrees of peripheral neuropathies, as measured by nerve conduction times in both sensory and motor nerves [195]. Chronic exposure to moisture-damaged buildings may also be responsible for respiratory tract infections, mucosal irritations, and asthma-like symptoms (the so-called Dampness and Mold Hypersensitivity Syndrome, DMHS). According to some authors, DHMS is a consequence of an immune system dysregulation that provokes hypersensitivities and enhanced sensitivity to infections [196]. Among the pathological consequences of DHMS, MCS is one of the most crucial candidates, as several symptoms are common in both conditions (see [65] for a comprehensive review). Hirvonen et al. [197] found an increase in the concentration of inflammatory cytokines in the nasal lavage fluid of people working in moldy polluted schools compared to controls, which provoked symptoms like catarrh, cough, rhinitis and fatigue.

Immunogenic dysregulation has also been observed following cytokine release due to exposure to xenobiotics [198,199]. Jalava et al. [200] found that fine particulate matter in urban air are triggers for inflammatory and cytotoxic effects. Moreover, Belpomme, Campagnac and Irigaray [201] observed that MCS and Electro-Hypersensitivity (EHS) patients exhibited higher levels of histamine, inflammatory processes affecting thalamus and limbic system, and oxidative stress compared to controls (for a consensus report on specific markers in electrohypersensitivity, see [202]). The authors considered these as reliable and objective etiopathogenic biomarkers for diagnosis, suggesting a common pathological mechanism in the two disorders [see De Luca et al. [34] for an interesting review on dependable biological markers of MCS and Hoover at al. [203] regarding the relationship between reliability of some immunological tests and MCS].

There are a number of studies whose results differ from the above. Grammer et al. [204] for example, observed that symptoms exhibited by a worker following formaldehyde inhalation were not caused by immunologically mediated asthma (see also [205]). Similarly, Fiedler et al. [2] found no abnormal immune system responses in their MCS patients. According to Mitchell et al. [206] immunological changes can be seen in the MCS population, but they are not very significant and do not affect all patients. Critical perspectives on the role of immunological involvement in MCS are provided by a number of authors [12,16,33,207,208].

### 8.5. Oxidative Stress Hypothesis

A number of theorists posit oxidative stress as a cause for MCS. The presence of elevated levels of nitric oxide (NO) and peroxynitrite (PN), two indices of oxidative stress, have been claimed to be an etiopathogenic marker in persons with this condition. Belpomme et al. [201,202] found in their MCS and EHS patients an increased amount of serum nytrotirosin, a marker of PN, suggesting the presence of oxidative stress. Similarly, Hirvonen et al. [198] observed an increased concentration of NO in their nasal lavage samples of persons exposed to mold-related microbes.

Perhaps the best evidence for a role of NO and PN as underlying biochemical mechanisms of MCS comes from a series of studies of Pall. According to Pall [209,210,211,212,213], hypersensitivity reported by MCS patients is likely due to a cascade of biochemical factors, which begins with an increase in N-methyl-d-aspartate (NMDA) receptor activation (e.g., due to organic solvents) which, in turn, give rise to an increase in NO and the oxidative product of PN (ONOO). Notably, environmental stressors that stimulate NMDA receptors also act on limbic kindling/neural sensitization and/or neurogenic inflammation processes. These provoke, as a final step, progressive extreme sensitivity to organic solvents and increased blood barrier permeability which increase the access of chemicals to the CNS and decrease natural detoxification processes.

Several classes of xenobiotics appear to lead to MCS, including hydrophobic organic solvents and numerous pesticides which trigger the biochemical NO/ONOO cycle. Furthermore, Pall [210,212] suggests that his biochemical interpretation of MCS symptomatology can be extended to other diseases like CFS, FM and Post Traumatic Stress Disorder (PTSD) These disorders share similarity to MCS and can be triggered by a common etiologic mechanism, namely elevated levels of NO/PN. Related studies that directly or indirectly support Pall’s hypothesis can be found in [34,199,214,215,216,217,218].

As Pall’s hypothesis appears to be linked to the limbic kindling/neural sensitization and neurogenic inflammation hypotheses, it potentially unifies several theories of the underlying mechanisms of MCS in a single interpretation [217]. Nevertheless, the degree to which Pall’s model can take into account and explain the multi-organ pathologies exhibited by MCS patients and the role of the multitude of triggering xenobiotics remains questionable.

### 8.6. Genetic Theories 

Numerous theories of MCS propose that some individuals are genetically predisposed to hypersensitivity. One concept is that MCS patients have fewer or less effective enzymes for detoxifying chemicals and for metabolizing drugs, thereby sensitizing them to the development of MCS symptoms. In accord with this concept, McKeown-Eyssen et al. [219] found, in a female cohort of 203 MCS cases and 162 controls, that the cases had higher levels of cytochrome P450 CYP2D6 (i.e., one of the most important enzymes involved in the metabolism of xenobiotics in the body) and n-acetyl transferase 2 (NAT2). The CYP2D isoform activates and inactivates toxins, whereas the NAT2 bioactivates acrylamines to protein-binding metabolites. Similarly, Schnakenberg et al. [220] compared genetic variants of several other genes involved in metabolizing a broad range of chemicals between 248 persons with marked self-reported chemical sensitivity to 273 who reported much less self-reported sensitivity. Those with the most self-reported sensitivity were more frequent carriers of variants of the glutathione S-transferase genes GSTM1, GSTT1 and GSTP1, as well as the NAT2 gene. Glutathione S-transferases are involved in detoxification processes and the protection of cells from oxidative stress.

As noted earlier, MCS has prominent features of panic disorder, a disorder for which the cholecystokinin B receptor (CCK-B) has been implicated [142]. In a CCK receptor allele typing study by Binkley et al. [143], 40.9% of 22 alleles from MCS patients were the CCK-B receptor allele 7, the same allele shown to be associated with panic disorder [221]. Only 9.2% of 22 alleles from matched normal controls were the CCK-B receptor allele Panic disorders are common in Gulf War Veterans who have been exposed to organophosphate pesticides, nerve agents, high concentrations of n,n-diethyl-m-toluamide (DEET) insect repellants, and toxic levels of pyridostigmine (a cholinergic agonist used to combat nerve agents). Haley, Billecke and La Du [222] found sick veterans with such symptom complexes were more likely to have the R allele (homozygous R or heterozygous QR) than the homozygous Q for the paraoxonase-1 (PON1) gene. This gene is associated with the protection of lipoproteins from oxidation and provides defense against a number of diseases, including cardiovascular disease. Low activity of the PON1 type Q arylesterase allozyme distinguished sick veterans from controls better than just the PON1 genotype or the activity levels of the type R arylesterase allozyme, total arylesterase, total paraoxonase, or butyrylcholinesterase. A history of advanced pyridostigmine acute toxicity was also correlated with low PON1 type Q arylesterase activity. These findings are in accord with the hypothesis that environmental chemical exposures can cause MCS-like neurologic symptoms in some Gulf War veterans and that genotype can impact the effect of such exposures.

A number of studies have not found clear genetic differences between MCS patients and controls. For example, Wiesmüller et al. [223], in a study of 59 MCS and 40 controls, found no significant allelic distribution differences of genetic polymorphisms in genes for the serotonin transporter (5HTT), superoxide dismutase 2 (SOD2), NAT1, NAT2, PON and PONDantoft et al. [224] compared expression levels of 26 genes involved in biochemical pathways proposed to be involved in MCS between 18 MCS patients and 18 controls, namely genes involved in immune regulation, sensory detection, the physiological stress response, and enzymes within the sphingosine-1-phosphate pathways. Assessments were performed both before and after the subjects were exposed to n-butanol for an hour in an environmental chamber. Similar gene expression levels were evident for the two groups both before and after the exposures. The MCS patients could not be differentiated from the controls based upon expression of the evaluated genes. More recently, Berg et al. [225] divided 96 MCS patients and 1207 controls into four MCS severity groups and genotyped them for variants of genes encoding cytochrome P450 2D6, NAT-2, PON1, methylene tetrahydrofolate reductase, and the CCK2 receptor. No statistically significant difference in frequency was found for any gene variant either between the patients and the controls from the population sample, or within the population sample between severity groups 1–4.

Although there are no genetic animal models of MCS, per se, it is known that mice with the gene-targeted deletion of the potassium Kv 1.3 channel have a 1000 to 10,000 fold lower detection threshold for odors, and increased ability to discriminate between odorants in comparison with their normal littermates [226], supporting the idea that hypersensitivity to odorants is possible via gene manipulations. To our knowledge, genes associated with hypersensitivity in rodents to odorants have not been explored in MCS cases although given the sparsity of measurable olfactory dysfunction in MCS such an endeavor may not be fruitful.

## 9. Assessment Procedures

An overall assessment of MCS and chemically intolerant patients can be made by means of quantitative inventories, questionnaires, and practical psychophysical olfactory tests. A number of such measures have been developed and validated and are available in the literature or commercially. They have proven useful in both clinical and experimental contexts in identifying MCS patients and quantifying their symptoms. 

### 9.1. Quantitative MCS Inventories and Questionnaires

Researchers and clinicians who participated at “the 1999 consensus meeting” in the United States to define diagnostic criteria for MCS [64] strongly suggested, as a first step in identifying MCS, the use of the inventory developed by Miller and Prihoda [66,67], the Environmental Exposure and Sensitivity Inventory (EESI; available online: http://www.chemicalsensitivityfoundation.org/chemical-sensitivity-questionnaire.html, accessed on 20 October 2021). This questionnaire is particularly relevant in detecting people sensitive to common chemical triggers. A shorter 50-item form of the ESSI, termed the Quick Environmental Exposure and Sensitivity Inventory (QEESI), has since been developed which is simpler and faster to use [66]. The QEESI was well received by the international scientific community and has been translated in various languages, resulting in the availability of several versions [10,227,228,229,230,231].

The QEESI is comprised of five sections, each of which contains 10 questions. The sections are: *Chemical Exposures, Other Exposures, Symptoms, Masking Index, and Impact of Sensitivities.* With the exception of the *Masking Index*, which requires yes or no answers, each section asks the subject to rate the degree to which they experience functional disability. In the case of *Chemical Exposures*, this relates to a series of different chemicals, whereas *Other Exposures* relates to exposures to various foods, drugs, fabrics, etc. Symptoms relates to various body parts, e.g., muscles, joints, stomach, heart, as well as mood and other psychological factors, whereas *Impact of Sensitivities* addresses how the symptoms influence everyday life (e.g., social relationships, travel, choice of clothing, use of personal care products). The *Masking Index* poses questions about personal habits, such as smoking, drinking, use of caffeinated beverages, exposures to chemicals in the home or workplace, etc. Separate scores are determined for each of the five QEESI sections, with normative data implying low, medium, and high chemical sensitivities.

Other questionnaires are also available to allow for the assessment of chemical intolerant patients. These include: (a) the Chemical Sensitivity Scale (CSS) and its short version (i.e., Chemical Sensitivity Scale for Sensory Hyperreactivity, CSS-SHR) for the assessment of self-reported negative affective reactions and behavioral problems following exposure to odorous and pungent chemicals [232,233]; (b) the Chemical Odor Sensitivity Scale (COSS) for assessing strong physical reactions to common environmental chemicals [234]; (c) the brief Chemical Odor Intolerance Index (CII) for evaluating the frequency of feeling ill after exposure to five chemicals [235]; (d) the University of Toronto Health Survey (UTHS) which identifies low-dose exposures symptoms [236]; (e) an unlabeled questionnaire devoted to measure chemical intolerance prevalence to 122 items representative of various chemical agents [109];(f) the Idiopathic Environmental Intolerance Symptom Inventory (IEISI) devoted mainly to the frequency of symptoms in MCS sufferers [237]; and (g) a German MCS-questionnaire [125] that assesses environmental agents that provoke MCS and related diseases. Independent of established diagnostic criteria, questionnaires and inventories are useful tools to identify people complainting about chemical intolerance [64].

### 9.2. Olfactory Tests

Given the large number of complaints of heightened olfactory function in MCS patients, it is prudent to obtain an overall quantitative assessment of their ability to smell, if only to dispel the notion of olfactory system involvement. In general, persons are inaccurate in self-assessing their smell function in the absence of total anosmia, so quantitative measures are needed to objectively assess such function. Over 50 quantitative olfactory tests are described in the literature, although only a few are commercially available. Most such tests are based upon the abilities to detect, identify, discriminate, and remember odor sensations (for reviews, see [238,239]).

The most widely used psychophysical olfactory test kits developed for clinical and experimental purposes are those developed at the University of Pennsylvania [239] and the University of Erlangen-Nürmberg [240]. All of these tests are commercially available. Those developed at the University of Pennsylvania are manufactured by Sensonics International, Haddon Heights, NJ, USA. These include the self-administered 40-item University of Pennsylvania Smell Identification Test (UPSIT^®^ [241], which has been published in 34 different languages and for which a revised 2020 version is available, as well as multiple versions of the self-administered 12-item Brief Smell Identification Test (B-SIT^®^; also known as the Cross-Cultural Smell Identification Test or CC-SIT [242]. Others are, the 8-item National Health and Nutrition Examination Survey (NHANES) odor identification test and the 12-item nonverbal odor memory/discrimination test (ODMT^®^). Both the UPSIT and B-SIT have been used in testing MCS patients [243]. The ODMT^®^ provides a score for each of three delay intervals (i.e., 10, 30 and 60 seconds; [242,244]). Also developed at the University of Pennsylvania is a sophisticated detection threshold tests which ascertain the lowest detectable concentration of a chemosensory stimulus an individual can detect, namely the Snap and Sniff^®^ Smell Threshold Test [239,245]. This test kit is comprised of 20 smell “wands”, five of which contain no odorant and the others half-log_10_ stimulus dilutions of phenyl ethyl alcohol ranging from 10^−2^ (strongest) to 10^–9^ (weakest) dilutions in United States Pharmacopeia (USP) grade light mineral oil. The odorant is contained within a rechargeable cartridge inside each wand. When the thumb of the operator pushes forward on a slide mechanism, a wick containing the odorant is exposed for sniffing within a housing that makes it impossible for the wick to directly touch the nose. Releasing the slide retracts the wick into the wand’s body. Reliability and normative data for these tests are available.

The Sniffin’ Sticks test kit [246,247,248], manufactured by Burghart Messtechnik, Hamburg, Germany, is a test of nasal chemosensory performance based on felt tip pen-like odor dispensing devices. It allows the assessment of odor thresholds using 16 different dilutions of n-butanol, prepared in a geometric series starting from a 4% n-butanol solution. More recent versions employ the odorant phenyl ethanol. In the odor discrimination task, 16 triplets of pens are administered in a randomized order, with two pens containing the same odorant and the third pen a different one. For each of the triplets, participants are requested to determine which of the three pens smells differently. Finally, odor identification is assessed with participants smelling 16 common odorants; identification of individual odors is performed selecting the appropriate label from lists of four descriptors. For a memory olfactory test based on the Sniffin’ Sticks kit, see [249]. The olfactory tasks can be hypothetically order on a continuum from the most sensorial to the most cognitive, namely: odor detection, intensity discrimination, quality discrimination, odor recognition, cued identification and free (non-cued) identification [249] and may be linked to each other in a hierarchical and parallel fashion [250]. A number of studies have examined MCS using Sniffin’ Sticks [75,79]. Reliability and normative data are available for these tests.

Sophisticated electrophysiology-based tests as well as computerized self-administered olfactory tests using olfactometers are available, although, largely because of cost and the need for trained technical support, they remain within the realm of research instruments and are not widely employed in MCS testing. Nonetheless, they are part of the armamentarium of olfactory test equipment available for those who wish to use them.

## 10. Conclusions and Future Directions

In this review, we have provided a synopsis of what is presently known about MCS, including its history, definition, demographics, prevalence, and etiology. Although no uniform set of standard medical tests has yet been identified that can explain the multitude of multisystem complaints of most MCS patients, careful medical evaluations have resulted in redirecting a number of such patients into more traditional clinical entities, including those involving respiratory, dermatologic, gastrointestinal, and psychologic/somatization disorders. Regarding the latter, advances in differentiating between organic and psychologic causes of the symptoms have been made. Significant advances also have been made in the development of valid questionnaires and inventories useful for quantifying and identifying complaint clusters of MCS symptoms that may ultimately be of value in establishing causal associations and guiding therapies. Despite the unorthodox complaint patterns and the complex interplay between biological and psychological elements of the involved patients, the patients clearly are suffering from the symptoms of MCS regardless of their cause. Controversy remains regarding whether many MCS-related symptoms precede or follow purported chemical exposures. The acceptance of MCS as a clinical entity by a wide range of governmental agencies can only facilitate a better understanding of the factors responsible for the symptoms, regardless of their apparent etiologies. Clearly, MCS, however defined, requires health care system attention and management within a wider rubric of understanding health in general relative to the social, political, and economic ecosystems responsible for optimizing health outcomes [11,18,56,70,251,252,253,254]. Not the least of such factors is air pollution. According to the World Health Organization, air pollution is among the top five risks for the development of chronic non-communicable disease, with the other risks being tobacco use, alcohol abuse, unhealthy diets, and physical inactivity. MCS is clearly intimately associated with such pollution.

In the future, more research is needed to accurately establish the specific chemicals responsible for MCS symptoms, as well as their relative potencies, and to better differentiate between psychological and organic causes via paradigms such as the one described by Zucco, Militello and Doty [8]. As noted by Rossi and Pitidis [9], carefully controlled longitudinal exposure studies are sorely needed to define offending agents and identify persons with difficulties in bioaccumulation and detoxification.

Importantly, more sophisticated imaging studies of MCS patients, such as PET studies using ligands to acetylcholinesterase and other neurochemicals potentially impacted by environmental toxicants, are needed to empirically define the influences of MCS on brain function. A number of authors have outlined needed study designs that have yet to be implemented to test specific hypotheses of illnesses induced by low levels of environmental chemicals (e.g., [63,191]), and it would seem prudent that future researchers carefully study their recommendations.

## Figures and Tables

**Table 1 brainsci-12-00046-t001:** Multiple chemical sensitivity prevalence rates among studies.

Authors	Year	Population	Estimated Prevalence (%)	Sample Size	Age (Yrs)	% Male	Medical Diagnosis or Self-Report *
Gyntelberg et al. [106]	1986	Denmark	12.5	50	NA **	86	Medical
Cullen et al. [57]	1987	US: Yale Occup clinic	1.8	2759	22–60	32	Medical
Bell et al. [107]	1993	US: Arizona	15	643	Mean: 18.6	21	Self-report
Kipen et al. [108]	1995	US; Hospital Waiting Rooms	5.5	705	28–66	31	Self-report
Lax and Henneberger [109]	1995	US: Syracuse NY	5.8	605	14% ≤ 35 71% 36–50 14% > 50	20	Medical
Bell et al. [88]	1996	US	15	809	19.1	40	Self-report
Meggs et al [110]	1996	US: Rural NC	33	1027	34% <40 37% 40–64 22% > 64	39	Self-report
Baldwin et al. [111]	1999	US: Arizona	23	181	15–50+	39	Self-report
Kipen at al. [112]]	1999	US: Gulf Registry Vets	13.1	1161	Mean: 35.2	91	Self-report
Kreutzer et al. [93]	1999	US: California	6.3	4046	>17	32	Medical
Kreutzer et al. [93]	1999	US: California	11.9	4046	>17	26	Self-report
Caress and Steinemann [113]	2003	US: National Sample	3.1	1582	<20 to >50	39	Medical
Caress and Steinemann [113]	2003	US: National Sample	12.6	1582	<20 to >50	39	Self-report
Haussteiner [114]	2005	Germany: Nat’l Sample	0.5	2032	>15	NR	Medical
Haussteiner [114]	2005	Germany: Nat’l Sample	9.5	2032	>15	NR	Self-report
Sears [18]	2007	Canada: Nat’s Sample	2.9	135.573	30+	NR	Medical
Park and Knudson [115]	2007	Canada: National pop	2.4	643	12 to >65	28	Medical
Berg et al. [116]	2008	Denmark	27	4242	18–69	46	Self-report
Fitzgerald [117]	2008	Australia: South Aust.	0.9	4009	18–75+	23	Self-report *
Caress and Steinemann [118]	2009	US: National Sample	11.6	1057	<20 to >50	35.4	Self-report
Steinemann [95]	2019	US; National Sample	12.8	1137	18–65	58	Medical
Steinemann [95]	2019	Australia: Nat’l Sample	6.5	1098	18–65	48	Medical
Steinemann [95]	2019	England: Nat’l Sample	6.6	1100	18–65	59	Medical
Steinemann [95]	2019	Sweden: Nat’l Sample	3.6	1100	18–65	60	Medical
Petersen et al. [119]	2020	Danish: Nat’l Sample	0.9	1590	44–63	41	Medical

* Based on question of whether a medical doctor has told them they have MCS or related disorders. ** Not available.

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
