# Peer review of "Multiple Chemical Sensitivity"

_brainsci, 2021, doi:10.3390/brainsci12010046_

Round 1

Reviewer 1 Report

This article reviewed Multiple Chemical Sensitivity (MCS) in terms of history, definition, demographics, prevalence, etiology and assessment. I think the authors provide a broad and detailed overview of MCS. However, I have some main concerns.

  1. The title of this article states that “it is state of the art”. However, most of the references are not that recent. In addition, it would be good that the authors present the number of publications per year on MCS.
  2. What’ the most important point or highlight in this review?
  3. The authors listed the existing biological and psychological etiology for MCS. I was wondering whether they have any new ideas or models contributing to this review? At least, they should present some future directions in this field.
  4. I think the topic in this article is suitable for this journal----Brain Sciences. As the authors demonstrated, MCS involved neurogenic inflammation and psychological symptoms. However, it seems that the authors do not describe enough brain sciences in the present paper. They should add more brain studies on MCS if they expect the paper to be published in this journal. For example, prediction processing models (e.g. Stephan et al., 2016) may provide some insightful perspectives.
  5. What’s the relationship between MCS and medically unexplained symptoms?
  6. What’s the role of symptom perception or interoception in MCS? Is it possible that the etiology of MUS involves in interoceptive dysfunction?
  7. The sensitivity to odorants is not well-defined in this article. Does the sensitivity means detecting odorants correctly or lower odor thresholds? Or, sensitivity means over-reactivity?
  8. In the “assessment procedure” part, the authors mention the olfactory tests. Could they also list some MCS studies or diagnoses which used the mentioned olfactory tests?
  9. I think the authors should add some up-to-date treatments for MCS?

I also have some minor concerns.

  1. There are quite a few long sentences in the article, which makes it difficult to understand. And I feel the logics in part 4 (olfactory system involvement in MCS) and “classical conditioning” part are difficult to follow.
  2. Please check the typo errors in the article. For example, in page 3, there only listed 4 definitional criteria for MCS by Cullen (1987), but there are five letters of an alphabet-----a, b, c, d, e, to indicate them.
  3. In page 12, the authors cited the paper of van den Bergh et al. (1999). However, I could not find it in the list of references in the end.
  4. In the last paragraph in page 12, the authors mention the “CC model”. What’s the full name?

Author Response

Please, see attachment.

Reviewer 2 Report

The review presented by Zucco and Doty provide an extensive overview of multiple chemical sensitivity. It was rather interesting to read through the history, definitions, and etiology underlying this condition.  

First, I think the title sort of leaves the reader hanging: Multiple Chemical Sensitivity: State of the Art. State of the Art what? It seems cut off to me. I would consider revising.

Next, the authors need to fine tune their writing and ensure this is written with one voice. Throughout the paper you can find examples of the word odor, spelled, odor and odour. Choose one and stick with it in the entire paper. Please re-read the manuscript for consistency. 

Abbreviations should be defined the first time they are used within the manuscript. Some abbreviations are never defined. Some abbreviations are defined multiple times or in different ways. The paper needs to be re-read thoroughly from beginning to end and correct these as they are a distraction from the content. Some examples are below:

  • in the abstract multiple chemical sensitivity is abbreviated MCS, but by the middle of the paragraph it is abbreviated MCI (which to me is mild cognitive impairment). This can also be found in the middle of page 9.
  • Page 11, the paragraph starting with "The presence of ..." needs work. Abbreviations and chemical formulas are used in this paragraph not seen before and are not defined. The grammar needs work. Sensitization is spelled with an s when elsewhere it is with a z (sensitisation). 
  • on page 13, second paragraph, the abbreviations CFS, SBS are used for the first time and never defined, whereas PTSD was used previously but defined here. 

Some sentences are worded awkwardly.

  • On the first page, for example, in the introduction, the sentence beginning with "In the first phase.." the commonly causing aversion phrase is awkward to read. I would consider rewording to "which commonly cause aversion"
  • On page 2, under the first History paragraph, midway through, commas are missing.
  • Page 7, the word complains should be complaints in the second to last line of the page. 
  • paragraph on page 11 referenced above. 

The references have varying spaces between them. This could just be a formatting issue when uploading the document, but it looks unprofessional. 

Also on the topic of references, in the American Academy of Allergy reference on page 2 (1981), I looked through the article and could not find anything in relation to accepting/not accepting MCS as a disease.

Additionally, in relation to various medical societies not accepting MCS as a disease, seeing these are all references from the 1980s-1990s, have any of these updated their opinions? We are now in 2021, and I feel like such organizations may have updated their stance on MCS. 

The table presenting the prevalence of Multiple Chemical Sensitivity could be made shorter or less extensive as it does not add that much to understanding prevalence. In addition, the paragraph preceding the table says that prevalence ranges from <1% to 37%. The highest prevalence presented in the table is 33%. 

What would really add to the effectiveness of the manuscript would be a proposed model or graphic that either shows symptoms and various diagnostic criteria or a model hypothesizing, based on the biological or psychological factors, of why/how multiple chemical sensitivity develops. 

Round 2

Reviewer 1 Report

The authors have addressed all my concerns with the previous version of this manuscript.